# Information about dissemination of trial results in patient information leaflets for clinicals trials in the UK and Ireland: The what and the when

**Matilda Bjorklund[1,2], Frances Shiely[1,3], Katie Gillies[2]***

**1** TRAMS (Trials Research and Methodologies Unit), HRB Clinical Research Facility, University College Cork, Cork, Ireland, **2** Health Services Research Unit, Health Sciences Building, Foresterhill, Aberdeen, United Kingdom, **3** School of Public Health, University College Cork, Cork, Ireland

* k.gillies@abdn.ac.uk

## Abstract

### Introduction

Complete and understandable information is vital for informed consent and this includes how and when potential participants can expect to receive trial results. Informing participants during informed consent about the sharing of trial results is important for addressing participants' needs, ensuring adherence to regulatory guidance, and in fulfilling a moral obligation.

### Methods

Patient Information Leaflets (PILs) were collated from across the UK and Ireland. Trial characteristics and data on disseminating trial results was extracted. Analysis included descriptive statistics and a directed content analysis approach. The content analysis framework was informed by regulatory guidance on PIL content and existing research on dissemination of trial results. Results were analysed using descriptive statistics and presented as a narrative summary as appropriate.

### Results

238 PILs from 178 trials were analysed. Of the 238 PILs, 74% (n = 176) provided information on sharing results with participants, 70% (n = 123) of which described passive methods of disseminating results that require active engagement from the trial participants, i.e., effort required by the participant to seek the results. The majority (90%) of PILs included more than one proposed mode of dissemination that largely targeted healthcare professionals rather than participants. Only 8% of PILs specified a time period for when results could be expected, 47% did not specify a time period (e.g. at end of trial), and 45% included no information on when trial results would be available.

### Conclusion

This study found that majority of the PILs included did include some information about dissemination of trial results. However, modes of dissemination tended to target researchers

**Data Availability Statement:** This manuscript reports a sub-study linked to a larger study that is generating an online repository of PILs. This repository (which includes all of the PILs included

in the analysis in this manuscript) will be made publicly available through www.trialforge.org on completion.

**Funding:** MB was funded for a summer period by the Health Research Board, Ireland through funding from the HRB Trials Methodology Research Network (Ref: HRB TMRN-2017-1). The Health Services Research Unit, Institute of Applied Health Sciences (University of Aberdeen), is core-funded by the Chief Scientist Office of the Scottish Government Health and Social Care Directorates (CZU/3/3). The funders had no involvement in study design, collection, analysis and interpretation of data, reporting or the decision to publish.

**Competing interests:** The authors have declared that no competing interests exist.

and clinicians rather than participants and information on when results would be available was often lacking. The findings highlight the need for further research that includes stakeholder input to identify what information on results summaries participants need at the point of making a decision about trial participation.

## Background

The World Medical Association Declaration of Helsinki specifies that potential participants should be *'adequately informed of the aims, methods, sources of funding, any possible conflicts of interest, institutional affiliations of the researcher, the anticipated benefits and potential risks of the study and the discomfort it may entail, post-study provisions and any other relevant aspects of the study.'* [ref] The provision of written information covering these core criteria is a central component of the process of seeking informed consent from potential trial participants. However, the evidence to support the inclusion of these items (and others) from a participants perspective is often lacking [1]. Recent evidence has demonstrated that potential trial participants do want to know about what will happen to the trial results when they are considering trial participation [2, 3]. One such study which focused on the relative importance of the UKs Health Research Authority (HRA)-specified information items in Patient Information Leaflets (PIL) for trials found that participants considered information on 'What will happen to the results of the study?' more important than 'Will my taking part in the study be kept confidential?' and 'What will happen if I don't want to carry on with the study?' [2].

For consent to be 'informed' it is necessary to ensure that potential participants are fully aware and understand what they should expect from the study they might take part in—this includes when and how they may receive the results of the study [4]. In the Consent and Participation Information Guidance from the UK's HRA it is clearly stated that one of the essential elements in a participant information sheet on which the information should be clear is "When and how it is anticipated that participants will find out the results of the study they are taking part in" [5]. Sufficient information in a way that is understandable to potential participants is vital throughout the whole process of the research from recruitment (for informed consent and choosing to participate) to the very end when results are disseminated to participants. A study on the impact of return of results on participants' views and expectations about their original decision to participate found that for some trial participants it highlighted that their original expectations had not been met [6]. Some participants felt betrayed by the staff involved in the trial with damage to trust, and that they had not fully understood the implications of taking part in the trial [6]. These findings highlight that setting expectations about receiving results at the point of making a decision about trial participation is an important part of the informed consent process.

It is clear that participants wish to know information about trial results at the point of consent. Therefore, informing participants about when and how trial results will be shared is important for addressing their needs, ensuring adherence to regulatory guidance, and delivering trials that are more participant centred in their design.

The purpose of this study was to examine a corpus of PILs to determine current practice with regard to sharing information about the 'what' and 'when' of trial results at the recruitment stage of clinical trials.

## Methods

### Data collection

PILs were collected from UK and Ireland based trials for a Health Research Board Trials Methodology Research Network funded study [7]. In brief, a request for PILs was made to all Irish Clinical Trials Units (CTUs) and all UK CTUs through publicly available contact details in October 2020 and supplemented with existing PIL datasets collated for existing projects led by co-applicants. There were no limits placed on the type of PIL sought as the authors sought a diversity of PILs in term of disease area, study population, trial phase etc. CTUs are specialists units with specific expertise in all aspects of design and delivery of clinical trials that often coordinate the delivery of several trials at any one time. This makes them a rich, and diverse, sample from which to source patient facing trial documents. PILs in the UK and Ireland are the written documents which detail the information required to support a participant to make an informed choice about trial participation and are similar to the 'consent forms' in US trials. They are detailed documents usually 15–20 pages in length.

### Data extraction and analysis

Various data on trial characteristics was extracted by a research assistant and overseen by FS. Information recorded was: study design; level of randomisation; phase of trial; commercial or non-commercial trial; organisation that approved the PIL; PIL version and date; study population; intervention and comparator; primary outcome; 'vulnerable' population (children, aged 16 and under, older adults, pregnant women, individuals with mental health conditions, retrospective recovered capacity consent/lack of capacity [8]); disease area; whether the PIL was a main trial PIL; permission for website; whether there was any mention of PPI involvement; and any description of sharing of trial results. For this study only the data on study design, vulnerable population, commercial or non-commercial trial, disease area, any mention of PPI involvement (at the dissemination stage only), and any description of sharing of trial results was used.

The data on sharing of trial results extracted from the PILs was analysed using a directed content analysis approach [9]. The content analysis framework was informed by a previous study on dissemination of trial results and guidance on the content to share in PILs in relation to study results [5, 10]. In addition to the a priori themes informed by existing guidance and research, inductive themes were developed during the coding process as follows: The codes were created inductively as more PILs were reviewed. The data were coded by one researcher (MB), and the coding was reviewed by FS and KG to increase the reliability of the coding framework. A random 10% checked by KG and any queries discussed amongst the research team. The meaning units and codes were reviewed and re-read with the original information from the PILs to ensure the information was captured sufficiently. Any information that was not relevant to the description of trial results was excluded. The codes were analysed and grouped into categories and sub-categories. This was done in a discursive online meeting with MB, FS and KG. The original text was again referred to at this stage of the analysis to ensure the researcher stayed close to the information provided in the PILs. Results are presented as a narrative summary as well as frequency counts and percentages for sample demographics and where relevant for trial results content data. The codes within the content analysis framework were: presence of information on dissemination of trial results; information on mode of delivery of trial results (i.e., in a medical journal, website, etc); whether information provision was active dissemination, passive dissemination or a combination (in terms of how the participants will acquire the information, i.e. active dissemination requires no action on behalf of the

participant such as a letter sent directly, whereas passive dissemination requires effort on their part such as having to locate results on a trial website or academic journal); when the results will be shared; information regarding participant anonymity; and whether any information on patient public involvement during results dissemination was included. Scientific meetings are differentiated from "conference presentations" in the coding as it was unclear what was meant by meetings, which could have included conferences but were separated for this study.

## Results

The sample demographics are summarised in Table 1. A total of 238 PILs were included from 178 trials; 30 trials provided multiple PILs that covered different age groups, parents, and legal representatives (n = 90, 38%). The funding of the trials was mostly non-commercial (n = 203, 85%) and a majority of the trials had a 2-arm design (n = 206, 86%). The most common disease area of study was oncology (n = 35, 15%), followed by obstetrics and gynaecology related trials (n = 20, 8%). Only four PILs mentioned PPI involvement during the results dissemination stage, three of which were from linked studies all conducted by the same research unit and within the same disease area.

Of the 238 PILs, 62 (26%) either contained no information on trial results provision or the information provided did not relate to results provision (for example, information on how participant data is stored). From the PILs which included information on trial results (n = 176, 74%), two discussed trial results without providing any information on how, or if, those results will be provided to the participants or any other interested parties (e.g., "As they are bloods for research purposes no results will be available to you"). The remaining 174 (73%) PILs that did provide information on whether study results will be provided were categorised as proposing either passive dissemination, active dissemination or a combination of both modes to disseminate trial results (Table 2). Well over half (n = 123, 71%) of the PILs described plans for active methods of disseminating results information and the remaining (n = 51, 29%) PILs included both active and passive dissemination mechanisms of sharing results information. There were no PILs that described only active dissemination methods of informing participants of the results.

Fig 1 illustrates the frequency of each of the active and passive modes of dissemination ordered from highest to lowest frequency. Scientific publication was the most frequently reported mode of dissemination with 166 (95%) of PILs that provided results information including it as one of the modes of dissemination. Content included examples such as "the results will be published in a medical journal" and "we hope to publish the results in a scientific journal". Other active modes of dissemination proposed in the PILs included: providing results in the public domain (i.e., trial website); participant initiated request; scientific meetings; and conference presentation. Scientific meetings included different variations of meeting including PILS stating "academic", "medical", "appropriate" and "maternity". The two most frequent modes of participant active dissemination were largely under-specified within the PIL. These include PILs stating in a variety of ways that participants will be provided with the results, when possible, without specifying how this will happen (e.g., "We will let you know the results of the study when it is finished unless you tell us that you do not wish to know" or "you will be provided with a summary of the findings at the end of the study"). Some of these descriptions gave the possibility of opting out of receiving results and others not, n = 17 (41%) and n = 24 (59%) respectively. Activemodes of dissemination proposed, but less frequently reported, included: community distribution (e.g., "They [people with study disease and their caretakers] will help us write up our findings . . . and share this information with the wider [study disease] community."); email; social media; postal; and patient PPI groups (e.g., "Findings from this

**Table 1. Sample characteristics.**

| Item | N (%) |
|---|---|
| Trials included | 178 |
| Trials with 2 or more PILs | 30 (17% of total trials) |
| PILs included in results information analysis | 238 (*221 from UK*) |
| PILs with no information on sharing of trial results | 62 (26% of PILs) |
| PILs that mention any PPI involvement at results dissemination stage | 4 (2% of PILs) |
| **PILs involving 'vulnerable' populations** *(children, aged 16 and under, older adults, pregnant women, individuals with mental health conditions, retrospective recovered capacity consent/lack of capacity)* | |
| Yes | 94 (40) |
| No | 141 (59) |
| Unclear | 3 (1) |
| **Funding source** | |
| Commercial trials | 10 (4) |
| Non-commercial trials | 203 (85) |
| Unclear | 25 (11) |
| **Trial design** | |
| 2 arm | 206 (87) |
| 3 arm | 15 (6) |
| 4 arm | 6 (2.5) |
| 5 arm | 2 (1) |
| 6 arm | 1 (0.5) |
| Unclear | 8 (3) |
| **Disease area** | |
| Oncology | 35 (14.7) |
| Infectious Diseases | 26 (10.9) |
| Respiratory | 22 (9.2) |
| Obstetrics & Gynaecology | 22 (9.2) |
| Neurology | 21 (8.8) |
| Urology | 21 (8.8) |
| Dermatology | 16 (6.7) |
| Cardiovascular | 12 (5.0) |
| Injury/trauma | 9 (3.8) |
| Gastroenterology | 7 (2.9) |
| Rheumatology | 6 (2.5) |
| Vascular | 5 (2.1) |
| Oral Health | 5 (2.1) |
| Mental Health | 5 (2.1) |
| Paediatrics | 4 (1.7) |
| Other/unclear | 22 (9.2) |

study will be . . . made available to patient groups/relevant charities"). One PIL reported that they 'may ask patients if there are any other methods they would prefer.' Demonstrating the study teams acknowledgment for the need to consider a one size fits all approach may not be appropriate. There were three PILs which were not possible to categorise as either passive or active dissemination due to the unclear wording; these PILs stated that "we will endeavour to inform participants and their GPs of the results, and any ensuing publicity" which does not

**Table 2. Inclusion of information on trial results summaries in PILs and means of dissemination.**

| | Information on results present in PIL | No information on result present in PIL |
|---|---|---|
| Passive dissemination only (i.e. effort required on part of participant) | 123 (70%) | - |
| Active dissemination only (i.e. no effort required on part of participant) | 0 (0%) | - |
| Both | 51 (29%) | - |
| Unclear/not reported | 2 (1%) | 62 (100%) |
| Total | **176** (100%) | **62** (100%) |

give clear information on how, or if, the results will be made available to participants or anyone else interested.

Of the 174 included PILs, 157 (90%) included more than one proposed mode of dissemination of trial results. One PIL (<1%) proposed eight different modes of dissemination and 17 PILs (10%) only included one mode. Roughly half (n = 91, 52%) of the included PILs had three different modes of dissemination. When considering who these multiple modes of dissemination are targeting (healthcare professionals/researchers or study participants) many of them are targeting healthcare professionals or researchers only rather than targeting study participants, but none are targeting participants alone. None of the most common combinations of modes of dissemination include wider communities as their targeted audience.

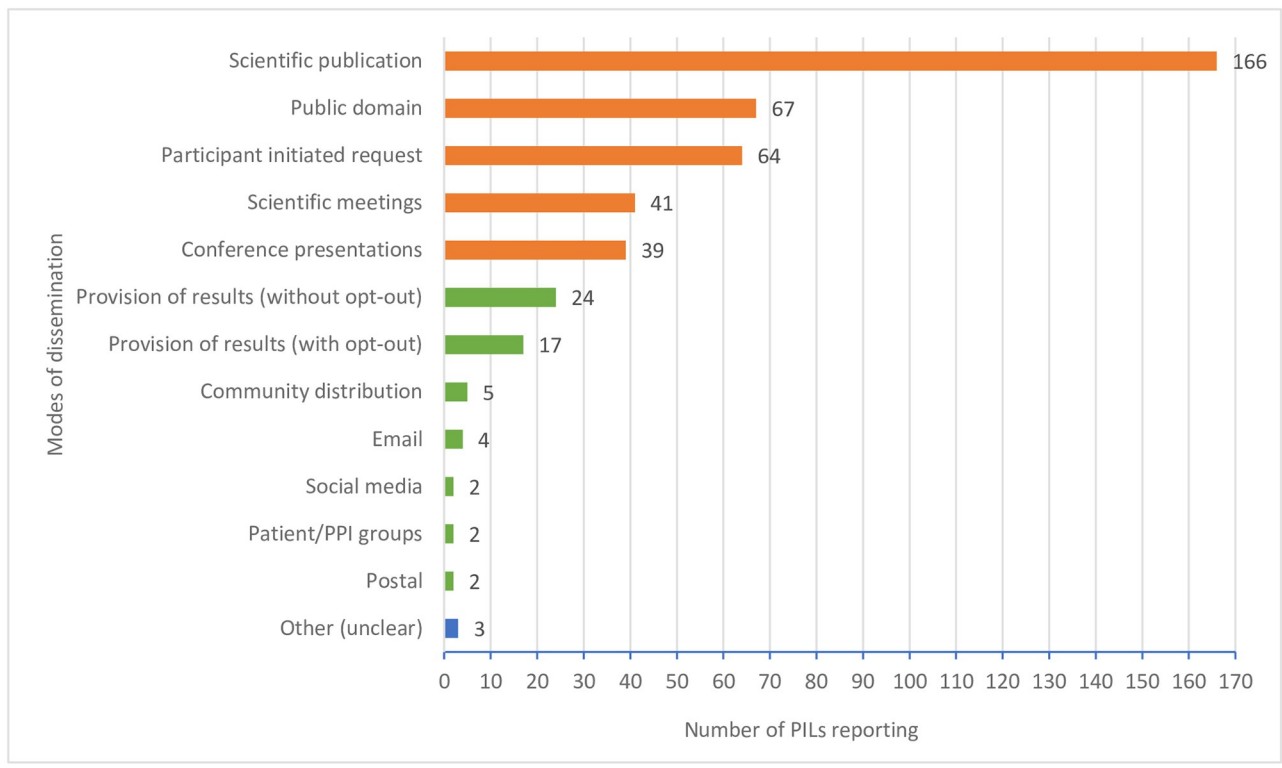

**Fig 1. Graph of active and passive modes of dissemination of trial results summaries.** (orange = active dissemination, green = passive dissemination, blue = unclear).

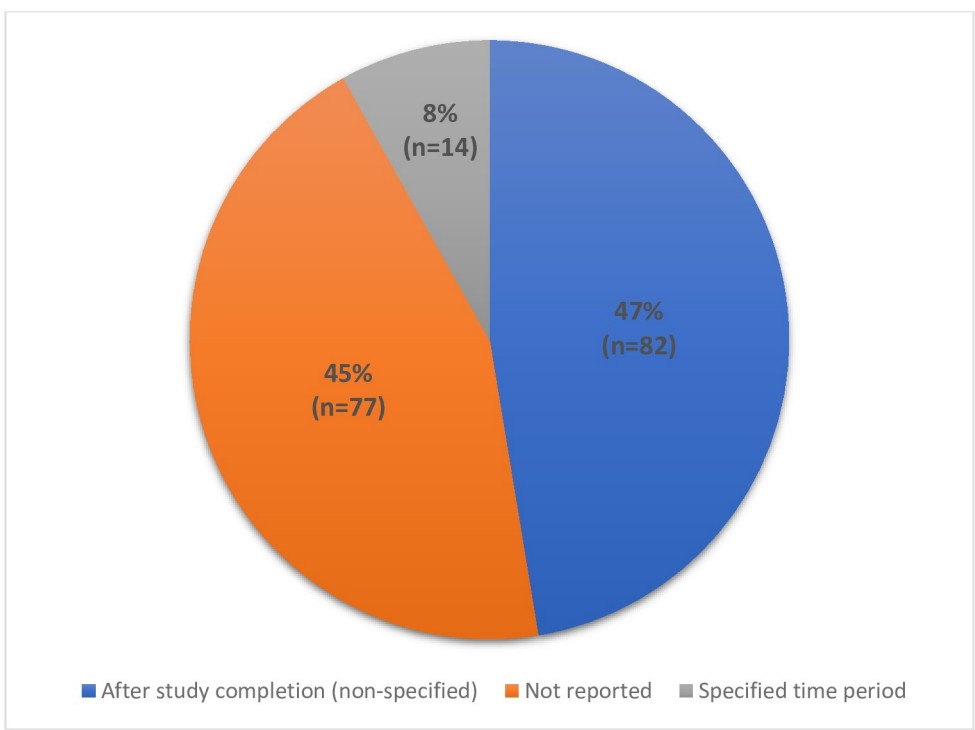

**Fig 2. Pie chart of information on timing of when results will be made available.**

Information contained in the PILs about when the trials results would be made available are illustrated in Fig 2. This was categorised as: after the trial has finished (unspecified time frame); at a specific time point or during a certain time period; or not reported. Almost half (n = 82, 47%) of the PILs with information on when the results would be available stated that this would happen at the end of the trial; this category also includes PILs which stated a variety of phrases that did not specify any time period but mentioned that it would happen after the trial is finished (for example, "when all patients have completed the study and the results are known" or "after the study has been completed"). The further 8% (n = 14) of PILs specified a time period when the results could be expected (e.g., "results are expected in 2019–2020"). The time periods ranged from six months after data collection up to at least 8 years (e.g., "the results are likely to be published in the six months following completion of the study", "this may not happen until 1 or 2 years after the study is completed" or "the results of this study are not likely to be available for at least 8 year"). However, 45% (n = 77) of the PILs did not report when the results would be made available.

Over half (n = 138, 58%) of the PILs did provide information on patient anonymity in relation to sharing of trial results summaries i.e., that no individuals will be identified in any reports or publications. The remaining 42% (n = 100) did not include any mention of patient anonymity within the information on dissemination of trial results.

## Discussion

This study has systematically analysed PILs with regard to how the practice of sharing trial results at the end of the trial is proposed and shared with participants during the informed consent process to participate in the trial. Nearly a third (n = 62, 26%) of the PILs in our

sample contained no information about whether, how or when trial participants could expect to receive the results of the trial they were considering participating in. The majority of PILs that included information on sharing of trial results, included active modes of dissemination, with dissemination through scientific publications being the most frequent mode of dissemination. Information on when results would be shared and patient anonymity was less frequently included.

The finding that nearly a third of our sample PILs did not include any information about how potential participants could find out about the results of the trial they are considering participating in is surprising. Existing UK guidance on development of PILs for research includes an item that states information on 'When and how it is anticipated that participants will find out the results of the study they are taking part in' should be included as an essential element of the study [5]. When paired with previous research that states that participants value information on 'What will happen to the results of the study' (scored as item 15 out of 32), there is a need to improve on this lack of attendance to participant preferences [2]. However, international guidance and legislation on informed consent does not make this information a requirement in PILs. For example, the Declaration of Helsinki highlights the need to make results publicly available but does not make this explicitly about access to participants nor does it state the need to include this information during the initial decision to participate [1]. In addition, ICH GCP states that potential participants should be informed that 'If the results of the trial are published, the subject's identity will remain confidential' but goes no further with regard to sharing information about trial results during informed consent [11]. As such, it may be that so see improvements in information content of PILs the necessary international guidance and legislation needs to reflect this.

One possible solution for creating PILs that are more relevant for potential participants is to include participant representatives in the development of the PILs and in particular to highlight in the PIL how patient involvement will or has been considered for dissemination of the results. Unfortunately, in the analysis of PILs presented in our study, we identified only four (2%) PILs mentioning the involvement of participants, or relevant population groups, in the results dissemination information within the PILs, but this may be a lack of reporting rather than a lack of doing. This is at odds with a previous audit of Integrated Research Application System (IRAS) applications which found that of those applications in which trial teams said they planned to share trial results with participants, almost 60% said they planned to include patient involvement in the dissemination stages [10]. This highlights a great opportunity to improve upon in future clinical trials when considering how information about results, and who is involved in their dissemination, should best be provided to participants when facing a decision about participation.

In addition to ensuring the content of PILs meets the needs and preferences of the potential participants, it is also important to consider preferences regarding the mode of delivery of trial results summaries. While it is obviously important to ensure the distribution of research results to the wider medical community, the need to make it accessible for participants who make this research possible should not be overlooked. The most common modes of dissemination, in combination with others and on their own, identified in this study were largely directed towards the wider healthcare research communities through scientific publications and conferences. This seems to present a disconnect in that most are not feasible for lay people to gain access to or understand. This finding resonates with recent research highlighting dissemination of trial results to participants is not commonplace irrespective of good intentions by trial teams [10, 12]. Furthermore, when dissemination does occur these results are often not shared in a way that is always appropriate for lay readers [12]. Similar to asking participants their preferences for mode of follow up, trial teams may consider moving to a position of

asking participants how they would like to receive the results of the trial or at the very least ensuring meaningful PPI during trial design and conduct to provide input into these questions.

Information on when then results of a clinical trial are ready to be published and more widely disseminated can be difficult to assess before the study has been completed. However, timing has been identified by a range stakeholders as an important factor when sharing clinical trial results with participants and so can be extended to being important to share during consent [13]. Nearly half of the PILs included in this study did not provide information on when participants could expect the results and a slightly larger portion of the PILs that did provide information did so in a way that did not provide any clear time frame. Only 8% of all PILs gave an indication on when results may be available. These results do not align with the HRA guidelines and highlights room for improvement. The importance and benefits of providing trial participants with the results has been previously established, outlined by the HRA guidelines, and now has co-designed evidence based guidance to support recommendations [13]. When considered alongside the upcoming requirements for trial results to be posted on a registry (or similar), providing more directive information and specifying when and how participants will have access to the results, at the point of consent, should be more commonplace.

## Strengths and weaknesses

This study builds on previous research on results information provision to clinical trial participants by adapting an existing framework for analysing how research teams stated they will provide information to trial participants through research and ethics approval applications [13]. Doing qualitative and iterative data analysis which builds on an existing framework helps in building the same knowledge base while also having the added benefit of further development for future research in the area. Furthermore, this study is more focused on the participants of clinical trials and how PILs share information on trial results which is important in that PILs are first and foremost intended for the potential participants of the trials during the informed consent process.

The majority of PILs (85%) analysed in this study were from non-commercial trials. This raises questions about the generalisability of the findings to commercially funded trials. Whilst some such trials were included in the analysis we did not conduct sub-group analyses to determine whether results were influences by source of funding. This would be an important focus for the future. Many of the trials included in this study were ongoing at the time of writing which means that the intentions for results dissemination stated in the PILs cannot be compared to what was actually enacted after the trial had concluded. It would be beneficial to be able to compare the intentions of research teams to what happens in practice and to build on that information by exploring trial participants preferences and perceptions of receiving the results in relation to their initial decision to participate.

## Conclusions

This study has analysed how PILs for clinical trials provide information for potential participants on the provision of the trial results. This study adds new insights to existing research on sharing trial results with participants by considering how this information on trial results is shared at the point of informed consent, when patients are considering their participation. The findings further highlight the variability in practice and supports the need for further research that included stakeholder input to identify what information on result summaries should be shared with participants at the point of informed consent to appropriately support informed choices about trial participation.

## Acknowledgments

Ellen Murphy and Genevieve Shiely Hayes for their contributions to data collection.

## Author Contributions

**Conceptualization:** Katie Gillies.

**Data curation:** Frances Shiely.

**Formal analysis:** Matilda Bjorklund, Frances Shiely, Katie Gillies.

**Funding acquisition:** Frances Shiely.

**Investigation:** Katie Gillies.

**Methodology:** Matilda Bjorklund, Frances Shiely, Katie Gillies.

**Project administration:** Matilda Bjorklund.

**Supervision:** Frances Shiely, Katie Gillies.

**Visualization:** Matilda Bjorklund.

**Writing – original draft:** Matilda Bjorklund, Katie Gillies.

**Writing – review & editing:** Matilda Bjorklund, Frances Shiely, Katie Gillies.

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
