## [Decision Letter · Decision Letter 0]

28 Feb 2022

PONE-D-21-29332Information about dissemination of trial results in patient information leaflets for clinicals trials: the what and the when.PLOS ONE

Dear Dr. Gillies,

Thank you for submitting your manuscript to PLOS ONE. After careful consideration, we feel that it has merit but does not fully meet PLOS ONE’s publication criteria as it currently stands. Therefore, we invite you to submit a revised version of the manuscript that addresses the points raised during the review process.

We look forward to receiving your revised manuscript.

Kind regards,

Ismaeel Yunusa, PharmD, PhD

Academic Editor

PLOS ONE

Journal Requirements:

Reviewers' comments:

Reviewer's Responses to Questions

**Comments to the Author**

1. Is the manuscript technically sound, and do the data support the conclusions?

Reviewer #1: Yes

Reviewer #2: Yes

2. Has the statistical analysis been performed appropriately and rigorously? 

Reviewer #1: Yes

Reviewer #2: Yes

3. Have the authors made all data underlying the findings in their manuscript fully available?

Reviewer #1: Yes

Reviewer #2: No

4. Is the manuscript presented in an intelligible fashion and written in standard English?

Reviewer #1: Yes

Reviewer #2: Yes

5. Review Comments to the Author

Reviewer #1: 1. Like Patient Information Leaflets (PIL), Health Research Authority needs to be abbreviated in the abstract.

2. Title is misleading about study area, study area needs to be made clear in the title i.e., UK and Ireland.

3. Line 178 does not make linguistic sense, it needs revision.

4. Line 220-222 and line 228-230 are an unusually long sentences that needs revision.

5. Authors alluded to the paucity of critical feedback to the participants in the PIL, what is the implication of this to ethics, PPI, and research governance? This needs to be highlighted.

6. Authors should suggest how this trial information should be deliver back to participants in a non-technical format.

7. Line 238 needs rephrasing.

8. Line 262-265 is long uninformative sentence with typos and nearly no break – punctuation. It needs revision.

Reviewer #2: PONE-D-21-29332

Information about dissemination of trial results in patient information leaflets 1 for clinicals trials: the what and the when

In the present study the authors analysed the content of information leaflets for clinical trials conducted in the UK and Ireland. Specifically, they analyse whether the Patient information Leaflets (PILs) contain information regarding when and how the trial results will be shared (either directly or indirectly with participants).

MAJOR POINTS

INTRODUCTION

The introduction makes the argument regarding the ethical obligation to return results and presents empirical evidence that patients would like this information. However, it isn’t clear what the specific motivating issue is for this study. On the one hand, there is an argument about consent only being informed if participants are “fully aware and understand what they should expect from the study they take part in” – yet this seems tangential to the present study as there is no patient data analysed (it may also be argued that there is a distinction to be made between what one requires to be disclosed vs understood, as Danielle Bromwich and others have argued).

The discussion about the essential content required by the HRA provides a closer link yet is only briefly discussed before moving on to participant views (again, less clear how this relates to the study or how the study informs that aspect).

What is missing is a problem statement or something more specific which notes the issue that the current study is addressing. Describing the content of PILS in terms of what results will be shared and when they will be shared doesn’t directly inform consent (it doesn’t say what should be disclosed) nor does it inform participant attitudes to that information. So it would be useful to understand why the present study was undertaken beyond simply describing the landscape (i.e. why was there a need to describe the landscape).For example, if this is an intermediary step to some form of consensus process to determine what information should be disclosed in this regard, that would be helpful.

METHODS

Data collection

For the data collection it would be helpful (especially to an international readership) to have more context on what a CTU is and their role. This would help understand why they were the source of the documents. At present it is unclear why a single entity might have access to many PILs.

It would be helpful to understand the legitimacy of the CTU to share specific trial information without liaising with the Principal Investigators of the trial. Was there discussion with the PI about releasing the materials to the authors? Or does the CTU have the right to do that – in effect are there any ownership issues relating to the PIL materials, or could they be considered in the public domain?

Also, I assume that the contact information for CTUs are publicly available (was there a specific contact at the CTU?)

This also leads into a potential limitation to be acknowledged (perhaps). Would the trials for which CTUs be engaged be publicly funded only, or could they be engaged in industry-funded trials? The latter may raise the issue of ownership (as mentioned above) but may also point to a potential selection issue if (and I am not sure this is or is not the case) there were differences in the content of information leaflets between publicly funded and industry funded trials (e.g. drug trials).

Can the authors also clarify what counts as a PIL here. For example, would that be a 20-page consent form for a drug trial, or would it be more of a pamphlet that may be supplementary to a formal consent form.

Were there any date limits placed around the trials? While it is noted that the request was made in October 2020 there is no information about whether there were inclusion or exclusion criteria set around the studies (were there dates of trial conduct? Did they have to be completed trials with results? Were they trials conducted in the UK and Ireland, or could they also be international trials? etc)

Data extraction and analysis

Why was only a subset of data used in the present study? Indeed, given the stated objective of determining “current practice with regard to sharing information about the ‘what’ and ‘when’ of trial results at the recruitment stage of clinical trials.” why was that information used?

It would be helpful to know more about the development of the coding framework; the HRA webpage cited as reference 1 provides no detail of content to be included (as far as I can tell).

While it is defined in the table 1, it would be useful to define ‘vulnerable population’ in the main text and the rationale for the categories included. For example, were these based on international guidelines about who should be considered vulnerable?

A number of codes within the content analysis framework are listed. Is this list a combination of those determined a priori and those that were inductively developed? If so, it would be helpful to distinguish them.

Was a particular framework used to categories the disease areas? Were health system intervention trials included?

Further, with the iterative addition of codes, were new codes then retrospectively applied to PILs that had already been coded prior to the addition of the new code? A little more detail on this process would be helpful.

To clarify, did MB apply a pre-existing framework only (i.e. deductive) or also inductively derive new codes? What training was undertaken prior to coding, was a training set used to ensure consistency of coding with the way the previous study was coded? This is important given that different individuals were involved and may have different perspectives.

Results

Please can I clarify the trial and PIL numbers: if 30/178 trials provided 94 leaflets, that should leave 148 leaflets (assuming 1 leaflet per trial) for 148+94 = 242 leaflets compared to the 238 leaflets reported (apologies if I have misunderstood).

In table 1 there is also variation in the denominator and whether this is the number of trials (N=178) or the number of PILs (N=238), which is somewhat confusing for the percentages.

Table 2 could perhaps be simplified as the 2nd column (‘No’) and 3rd column (No information) are essentially just categories such that the list of categories would be:

• Yes: Participant active only (i.e. effort required on part of participant)

• Yes: Participant passive only (i.e. no effort required on part of participant)

• Yes: Both

• No: Explicit statement of no dissemination of finding

• Unclear/not reported

This would make the summary statistics (%) neater

Was there any consideration regarding assessments of whether the leaflets were consistent with /adherent to the HRA guidance? Would it be possible to quantify that beyond those that simply did not provide information about results?

Was any consideration made regarding factors that might be associated with better reporting of trial result availability? That would seem useful information in terms of understanding if there are areas of good practice.

DISCUSSION

The authors note that:

“The finding that nearly a third of our sample PILs did not include any information about how potential participants could find out about the results of the trial they are considering participating in is surprising given existing guidance on development of PILs for research; this includes an item that states information on ‘When and how it is anticipated that participants will find out the results of the study they are taking part in’ should be included as an essential element of the study.”

While I take the point – and the issues is likely one of a generalisable nature – it should be clarified that this is HRA guidance (if I am correct) and, given that some leaflets were from CTUs in Ireland does this apply? (i.e. does the HRA guidance apply to Ireland as a regulatory feature or would there be separate guidance in Ireland that may not have the same requirement). I note there is no breakdown between leaflets for trials in the UK vs Ireland.

It would also be useful to note if the timing of results is important information (i.e. is the HRA guidance based on evidence or why this specific piece of information is key)

I also wonder on the issue about the timeframe for making results available. In some jurisdictions it is common practice to comment on the expected timeframes of a study, although this may be separate to explicit information about when results would be available. I wonder to what extent isolating this information about trial results from the rest of the PIL prevents that broader contextual understanding (e.g. if a PIL says elsewhere that the trial is expected to be completed in 2 years, is it reasonable to infer from this that the results would be available some period after 2 years).

An issue not discussed, and which may change with time, is the requirement to post trial results in a registry. Can the authors reflect on the effect of legislation requiring authors to make results available in registries?

MINOR POINTS

Several of the acronyms are not explained in their first instance and may not be familiar to an international audience. For example, what are CTUs or the HRB TMRN-TMRP?

In the abstract it appears that the number of leaflets is missing in several of the parentheses.

A minor suggestion – and this admittedly might be a personal take so is for consideration only – is the orientation of the terms “active participation” and “passive participation”. Given it is the responsibility of the researcher to disseminate the findings I wonder if it might be more instructive (and perhaps consistent with the idea of responsibility) to have labelled the “active participation” as “passive dissemination” – thereby noting that it is a lack of activity on the researchers behalf. This would also be consistent with terminology around Knowledge Translation and the passive dissemination (publishing) and more active forms of dissemination. Thus “passive participation” would become “active dissemination”, again to convey that the researchers are taking active steps to disseminate the trial results. As I say, this is for consideration only.

6. PLOS authors have the option to publish the peer review history of their article (what does this mean?). If published, this will include your full peer review and any attached files.

Reviewer #1: No

Reviewer #2: **Yes: **Stuart Nicholls

---

## [Editor Report · Decision Letter 1]

11 May 2022

Information about dissemination of trial results in patient information leaflets for clinicals trials in the UK and Ireland: the what and the when.

PONE-D-21-29332R1

Dear Dr. Gillies,

We’re pleased to inform you that your manuscript has been judged scientifically suitable for publication and will be formally accepted for publication once it meets all outstanding technical requirements.

Kind regards,

Ismaeel Yunusa, PharmD, PhD

Academic Editor

PLOS ONE
---

## [Editor Report · Acceptance letter]

13 May 2022

PONE-D-21-29332R1 

Information about dissemination of trial results in patient information leaflets for clinicals trials in the UK and Ireland: the what and the when. 

Dear Dr. Gillies:

I'm pleased to inform you that your manuscript has been deemed suitable for publication in PLOS ONE. Congratulations! Your manuscript is now with our production department. 

Kind regards, 

on behalf of

Dr. Ismaeel Yunusa 

Academic Editor

PLOS ONE